# PRISM: Agentic Retrieval with LLMs for Multi-Hop Question Answering

## Abstract

Retrieval plays a central role in multi-hop question answering (QA), where answering complex questions requires gathering multiple pieces of evidence. We introduce an Agentic Retrieval System that leverages large language models (LLMs) in a structured loop to retrieve relevant evidence with high precision and recall. Our framework consists of three specialized agents: a Question Analyzer that decomposes a multi-hop question into sub-questions, a Selector that identifies the most relevant context for each sub-question (focusing on precision), and an Adder that brings in any missing evidence (focusing on recall). The iterative interaction between Selector and Adder yields a compact yet comprehensive set of supporting passages. In particular, it achieves higher retrieval accuracy while filtering out distracting content, enabling downstream QA models to surpass full-context answer accuracy while relying on significantly less irrelevant information. Experiments on four multi-hop QA benchmarks—HotpotQA, 2WikiMultiHopQA, MuSiQue, and MultiHopRAG—demonstrates that our approach consistently outperforms strong baselines.

## 1 Introduction

Retrieval systems lie at the heart of question answering, yet designing them remains highly challenging (Asai et al., 2023). Their effectiveness depends on striking the right balance between precision—ensuring that retrieved passages are relevant and free from distracting noise—and recall—guaranteeing that no essential evidence is left behind. This trade-off is especially critical in multi-hop reasoning, where the reasoning chain spans multiple passages: missing even one can break the chain, while excessive irrelevant context can obscure the signal and degrade performance. With the advent of large language models (LLMs), these challenges are amplified. LLMs struggle with the *lost-in-the-middle* phenomenon, overlooking crucial evidence buried in long contexts (Liu et al., 2024), and are prone to hallucination when information is incomplete or noisy (Laban et al., 2024). Thus, enhancing retrieval to provide compact, comprehensive, and faithful evidence is essential for reliable reasoning in LLM-based systems.

These challenges are particularly acute for complex questions requiring multi-hop reasoning, where evidence must be drawn from multiple documents and combined into a coherent chain (Yang et al., 2018; Ho et al., 2020; Trivedi et al., 2022). For example, answering "Which painter who shared a house with Vincent van Gogh was married to a Danish ceramist?" entails finding who shared a house with van Gogh (first hop), then using that result to find who that person married (second hop). Solving such queries is challenging because a system must retrieve the relevant evidence for each reasoning step while ignoring the many distractors present in a large corpus like Wikipedia.

A rich line of research has explored retrieval for multi-hop question answering, yet important gaps remain. Early pipeline approaches such as GraphRetriever and Multi-hop Dense Retrieval (Asai et al., 2020; Xiong et al., 2021; Qi et al., 2019) relied on iterative query rewriting or entity linking to follow reasoning chains. While effective in controlled settings, these methods suffered from severe error propagation—mistakes in early hops irreversibly degraded final performance. Later frameworks such as ReAct (Yao et al., 2023) and IRCoT (Trivedi et al., 2023a) coupled large language models with retrieval, interleaving chain-of-thought (Wei et al., 2022) reasoning and evidence gathering. These approaches improved recall by letting intermediate reasoning guide the search, but

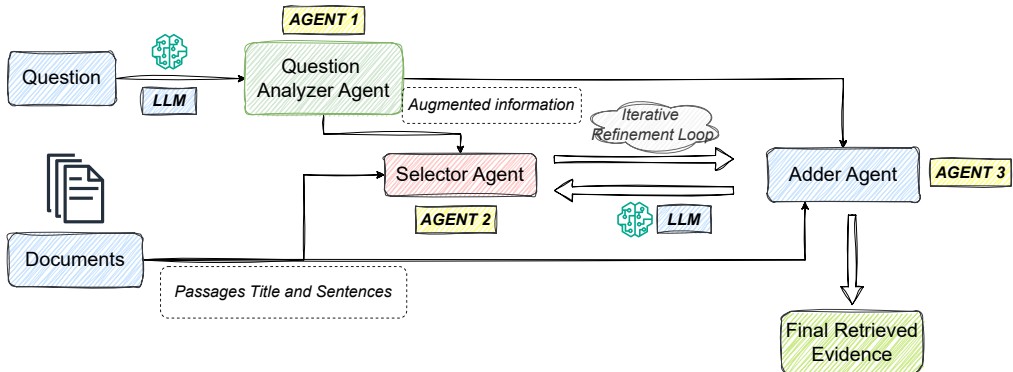

Figure 1: Overview of Agentic Retrieval Framework (PRISM). The complex question is decomposed by the Analyzer into sub-questions, the Selector narrows down relevant evidence for precision, and the Adder expands context for recall. The loop iterates $N$ times to produce a refined evidence set for QA.

often at the cost of large, noisy evidence sets, since precision mechanisms were limited. IRCoT in particular emphasizes recall, leading to distractor-heavy contexts that can obscure reasoning.

Complementary efforts have aimed to improve relevance and coverage through reranking, as in RankZephyr (Pradeep et al., 2023). Likewise, set-wise selection (Lee et al., 2025) suggests that moving beyond naive top-$k$ retrieval—by incorporating reasoning, selection, and refinement—is crucial for complex reasoning. However, these single-pass strategies are limited in that if essential evidence is absent from the initial retrieval pool, there is no mechanism for recovering it. Pruning-based methods such as Provence (Chirkova et al., 2025) improve efficiency by discarding irrelevant passages, but risk harming recall when essential evidence is mistakenly removed. Iterative self-refinement methods (e.g., Self-RAG (Asai et al., 2023)) attempt to balance these trade-offs, but lack a principled way to separate precision from recall, often generating redundant or hallucinated context.

Despite these advances, a key challenge remains: constructing evidence sets that are both *comprehensive and concise*. Standard retrievers and rerankers typically return a top-$k$ list of individually relevant passages, but the set as a whole is often redundant or incomplete. More broadly, retrieval-augmented generation faces fundamental limitations: (i) single-vector embeddings cannot represent all possible query–document relevance patterns as candidate sets grow combinatorially (Weller et al., 2025); (ii) long-context LLMs systematically overlook mid-context evidence (Liu et al., 2024); and (iii) irrelevant material in long contexts increases hallucination (Laban et al., 2024). Together, these findings highlight that simply scaling embeddings or extending context windows cannot resolve the precision–recall trade-off.

In this work, we introduce **PRISM** (Precision–Recall Iterative Selection Mechanism), an agentic retrieval framework that explicitly separates precision-oriented filtering from recall-oriented addition in an iterative loop. Concretely, PRISM employs three collaborating agents: a **Question Analyzer** that decomposes the question into sub-questions targeting the required facts; a **Selector** that filters the candidate pool to eliminate distractors; and an **Adder** that reconsiders unselected candidates to recover any overlooked evidence. The Selector and Adder iterate, refining the evidence set until it is both compact and complete.

Our contributions are threefold. **(1)** We propose **PRISM**, an agentic retrieval framework that explicitly separates precision-oriented filtering from recall-oriented addition, enabling fine-grained control over the precision–recall trade-off in multi-hop QA. **(2)** We show that PRISM consistently produces evidence sets that are both compact and complete, achieving higher precision and recall than recent baselines on HotpotQA, 2WikiMultiHopQA, MuSiQue, and MultiHopRAG. **(3)** We demonstrate that these improved evidence sets translate into stronger end-to-end QA, allowing LLM readers to match or surpass full-context performance, with notable gains on the most challenging multi-hop benchmarks where distractors typically hinder accuracy.

## 2 METHODOLOGY

We propose **PRISM** (Precision–Recall Iterative Selection Mechanism), an agentic retrieval framework that decomposes the multi-hop evidence gathering process into a sequence of coordinated steps. PRISM employs three LLM-based agents with distinct roles: the **Question Analyzer**, which decomposes complex queries into sub-questions; the **Selector**, which filters candidate evidence to maximize precision; and the **Adder**, which supplements missing evidence to improve recall. Each agent is instantiated as a prompted LLM with instructions tailored to its task. Operating in an iterative loop, these agents collectively produce a compact yet comprehensive set of supporting passages. Figure 1 illustrates the overall architecture and data flow. We provide an illustrative example in the Appendix A.5.

### 2.1 QUESTION ANALYZER AGENT

Multi-hop questions often intertwine multiple entities, relations, and constraints into a single complex query, making it easy for retrievers or LLMs to chase partial matches or irrelevant context. The Question Analyzer mitigates this risk by decomposing the query into a structured set of sub-questions that make the evidence requirements explicit and define a focused search space for later agents (Fu et al., 2021). This decomposition reduces the chance of missing critical hops while preventing downstream stages from being overwhelmed with loosely relevant candidates. Concretely, we prompt an LLM with a decomposition template that encourages explicit reasoning, and it returns a list of sub-questions, each targeting a distinct factual unit required to answer the original query.

### 2.2 SELECTOR AGENT

The Selector serves as a precision-focused filter that removes distractors introduced during retrieval. Large corpora inevitably yield passages that share surface-level words with the query but are semantically off-topic, and passing such noise directly to a QA model leads to error propagation and wasted context budget. The Selector mitigates this by enforcing a high bar for inclusion: it anchors the reasoning chain with the relevant passages that are strongly aligned with the sub-questions.

Concretely, the agent is prompted with detailed instructions, the sub-questions, and the retrieved passages, and tasked with outputting only those that directly support the query. This filtering stage yields evidence sets with very high precision, ensuring that downstream reasoning operates over reliable, compact inputs and mitigating hallucinations as well as lost-in-the-middle effects. The trade-off, however, is that some relevant passages may be excluded if their connection is less explicit—a limitation addressed by the Adder agent in the next stage.

### 2.3 ADDER AGENT

While precision is vital, overly strict filtering risks omitting complementary or bridging facts that are crucial for multi-hop reasoning. The Adder is introduced to counterbalance this risk by explicitly prioritizing recall. Its role is to audit what the Selector left behind and add evidence that fill logical gaps—such as a spouse relation that pairs with a shared-house relation, or a bridging entity connecting two documents. This two-phase design, ensures that the final evidence set is both compact and complete. The instructions for the Adder agent are similar to the Selector's, but with a different emphasis. We provide it with the same list of candidates, the sub-questions, and additionally indicate which evidence have already been selected by the Selector.

In essence, the Adder is a second pass through the candidates, but primed to look for anything important that was left out. One common scenario is when the answer to the sub-question actually requires combining two facts from different passages. The Selector might pick one of them (e.g. a passage that gives part of the answer), not realizing another evidence from different passage is also needed for the full answer. The Adder, seeing what's selected and what the question is, can identify the complementary piece. In the ideal outcome, after the Adder, the union of selected + added passages equals the complete set of supporting documents needed for answering the question. If the Selector already did a perfect job, the Adder will simply output nothing new. But if something was missing, the Adder ensures recall goes up. This two-phase selection is analogous to having a

strict filter followed by a gentle filter: one says "only keep absolutely sure things", the next says "don't leave out anything that might be required".

## 2.4 ITERATION AND MULTI-HOP HANDLING

Multi-hop questions typically require reasoning over multiple intermediate facts. After decomposition by the Question Analyzer, we provide the original question and all generated sub-questions together to the Selector and Adder agents. This joint formulation allows the Selector⇔Adder Cycle to identify the minimal set of evidence that collectively cover the reasoning chain.

The Selector⇔Adder cycle is repeated for several iterations, with the evidence set refined at each step. At the end of the process, evidences from all iterations are merged and de-duplicated, producing a compact set that typically contains only the essential evidence required for the multi-hop question while maintaining high precision and recall. This final set is then passed to a Answer Generator agent.

## 2.5 ANSWER GENERATOR AGENT

To demonstrate the overall question answering performance of our retrieval framework we introduce a **Answer Generator** agent, which produces the answer given the retrieved evidence set using our proposed framework. After the Question Analyzer, Selector, and Adder agents collaboratively construct a compact but comprehensive set of supporting evidence, we provide this context along with the original question to a large language model instructed to generate the final answer. We implement the Answer Generator as a prompted LLM operating in a **zero-shot** setting, without task-specific fine-tuning. This design choice allows us to directly assess how improvements in retrieval quality translate into downstream QA performance, while avoiding additional supervision or domain adaptation.

## 3 EXPERIMENTAL SETUP

### 3.1 DATASETS

We evaluate our framework on three standard multi-hop QA benchmarks in the open-domain setting, where supporting passage and/or facts can be retrieved from a corpus, and additionally include the MultiHop-RAG (Tang & Yang, 2024) dataset for specialized retrieval analysis. **HotpotQA** (Yang et al., 2018) contains questions requiring reasoning over multiple paragraphs, and we use its open-domain variant in which systems must retrieve relevant paragraphs from the full Wikipedia index, with supporting sentences provided for evaluation. **2WikiMultiHopQA** (Ho et al., 2020) features questions that connect two Wikipedia pages via shared entities; following prior work, we employ only the structured text component using its development set. **MuSiQue** (Trivedi et al., 2022) consists of questions requiring 2–4 reasoning hops and was specifically constructed to prevent reasoning shortcuts; we use the answerable subset defined by Trivedi et al. (2023a). **MultiHop-RAG** (Tang & Yang, 2024), a recently proposed benchmark in the RAG (Retrieval-Augmented Generation) domain, introduces multi-hop queries over a news-article knowledge base; we use this dataset particularly to assess the retrieval precision and recall of evidence selection in our framework. [1]

### 3.2 BASELINES AND EVALUATION METRICS

**Baselines.** We compare our Agentic Retrieval framework against several strong baselines. **IR-CoT** (Trivedi et al., 2023a) couples chain-of-thought reasoning with iterative retrieval. **SetR** (Lee et al., 2025) performs set-wise reranking with LLM reasoning, which we approximate by instracting LLMs to select a subset from the top-20 retrieved passages in a single step. **Oracle Gold** uses gold supporting paragraphs provided by the datasets, representing an upper bound for retrieval and QA. We also report a **No Retrieval (Full Context)** baseline, where LLMs answer questions using full context, and compare our results against DSP (Khattab et al., 2022), DecomP (Khot et al., 2023), RankZephyr (Pradeep et al., 2023), and RankGPT (Sun et al., 2023).

---

[1]For evaluation, we sample 500 instances from each dataset, constrained by computational budget, while ensuring that the subset remains representative of the overall distribution.

**Evaluation Metrics.** We assess retrieval using standard supporting fact metrics, treating each passage as the retrieval unit. A passage is correct if it contains at least one gold supporting sentence (HotpotQA, 2WikiMultihopQA) or matches a labeled supporting paragraph (MuSiQue, MultiHopRAG). We report **Precision** (fraction of retrieved passages that are gold) and **Recall** (fraction of gold passages retrieved), and also track the number of retrieved passages, as compact evidence sets improve efficiency and reduce noise. For HotpotQA and 2WikiMultihopQA, we additionally report **fact-wise precision and recall** at the sentence level. End-to-end QA is evaluated with **Exact Match (EM)** and token-level **F1**. We further measure performance under three conditions: (i) *full context*, where the QA model sees all retrieved passages including distractors, (ii) *retrieved evidence only*, using the filtered set from our framework, and (iii) *gold evidence only*, where the QA agent receives only the labeled supporting facts. This setup enables us to evaluate both retrieval quality and its effect on QA accuracy.

### 3.3 IMPLEMENTATION DETAILS

We implement all agents as prompted large language models. Unless stated otherwise, we use GPT-4o (Hurst et al., 2024), Gemini-2.0-Flash-Lite(Team, 2025) and DeepSeek-Chat(DeepSeek-AI et al., 2024), all state-of-the-art models with large context windows, enabling evaluation over all candidate passages. Prompting is performed in a zero-shot setting, without task-specific fine-tuning. We enforce consistent structured outputs (lists of indices/titles) to ensure reliable parsing. The Selector⇔Adder cycle is repeated at most $N = 3$ iterations, keeping the number of LLM calls tractable. As shown in Section 4, the system achieves strong retrieval performance, which directly translates into higher Question Answering accuracy.

### 4 RESULTS AND DISCUSSION

### 4.1 MAIN RESULT

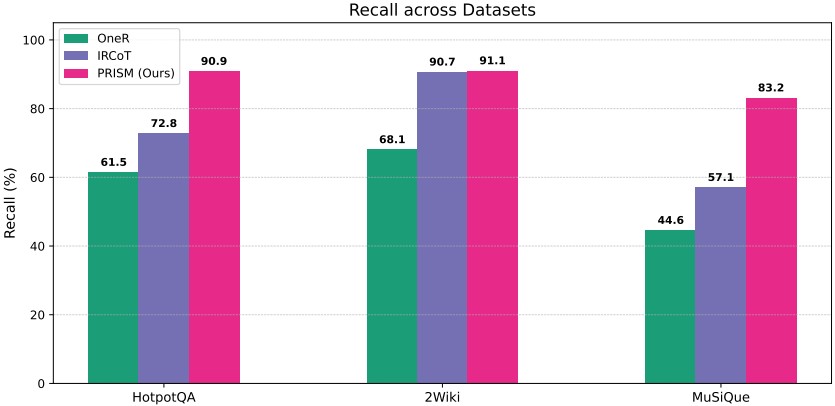

Figure 2: Passage recall across HotpotQA, 2WikiMultiHopQA, and MuSiQue. Our agentic retrieval framework (PRISM) consistently outperforms OneR and IRCoT (Trivedi et al., 2023b), with especially large gains on the challenging MuSiQue benchmark.

**Retrieval Performance.** Figure 2 reports recall performance across HotpotQA, 2WikiMultihopQA, and MuSiQue dataset. Our agentic retrieval framework achieves consistently higher recall than both OneR (one pass retriever), which utilize BM25 (Robertson & Zaragoza, 2009) to retrieves $K$ paragraphs using the original question as a single query, and IRCoT, which guides retrieval through Chain-of-Thought reasoning, across all benchmarks (Trivedi et al., 2023b). On HotpotQA, our method attains **90.9%** recall compared to 61.5% for OneR and 72.8% for IRCoT. On 2WikiMultihopQA, we reach **91.1%** recall, improving over OneR (68.1%) and slightly exceeding IRCoT (90.7%). The largest margin appears on MuSiQue, where our approach achieves **83.2%** recall versus 44.6% and 57.1% for OneR and IRCoT, respectively. These gains validate the importance of explicitly balancing precision and recall through our Selector⇔Adder loop: the Selector removes distractors while the Adder recovers missed but necessary passages, ensuring a comprehensive evidence set.

Table 1: Retrieval performance on the MultiHopRAG dataset. Metrics are precision (P) and recall (R). Our agentic retrieval framework substantially outperforms all baselines, achieving the highest recall while maintaining competitive precision.

| Method | Precision (P) | Recall (R) |
|---|---|---|
| BM25 (Robertson & Zaragoza, 2009) | 11.09 | 24.13 |
| bge-large-en-v1.5 (Xiao et al., 2023) | 16.12 | 32.32 |
| RankGPT (GPT-4o) (Sun et al., 2023) | 17.99 | 36.01 |
| SETR-CoT & IRI (Lee et al., 2025) | 22.68 | 36.69 |
| **Agentic Retrieval (PRISM)** | **28.18** | **42.22** |

We also assessed our proposed framework's retrieval performance using the MultiHopRAG dataset (Tang & Yang, 2024) following Lee et al. (2025). Table 1 summarizes retrieval results on the MultiHopRAG dataset. Our agentic retrieval framework achieves a precision of 28.18 and a recall of 42.22, which represents a large improvement over all baselines. Classical BM25 (Robertson & Zaragoza, 2009) performs poorly with only 11.09 precision and 24.13 recall, while dense retrievers such as bge-large-en-v1.5 (Xiao et al., 2023) and RankGPT (Sun et al., 2023) improve recall to around 32–36 but still fall significantly short of our method. SETR-CoT & IRI (Lee et al., 2025) narrows the gap with 22.68 precision and 36.69 recall, yet our framework surpasses it by about 6 points in recall and 4 points in precision. These gains highlight that the combination of precision-oriented selection and recall-oriented addition in our agentic loop is also effective for the MultiHopRAG dataset, which demands accurate recovery of multiple supporting evidence across hops.

**Fact-Level Retrieval Performance.** To provide a more fine-grained evaluation of retrieval quality, we also report sentence-level (fact-wise) precision, recall, and F1 for our agentic retrieval framework on HotpotQA and 2WikiMultiHopQA. Unlike passage-level metrics, which count an entire paragraph as correct if it contains at least one gold supporting fact, the sentence-level evaluation requires retrieving the exact gold supporting sentences.

Table 2: Fact-level retrieval performance of our agentic retrieval framework on HotpotQA and 2WikiMultiHopQA. Metrics are precision (P), recall (R), and F1 at the fact (sentence) level.

| Dataset | P | R | F1 |
|---|---|---|---|
| HotpotQA | 56.70 | 75.51 | 64.77 |
| 2WikiMultiHopQA | 60.81 | 74.42 | 66.93 |

These results show that our framework not only retrieves the correct passages but also identifies the exact supporting facts needed to answer multi-hop questions. This highlights the ability of the Selector⇔Adder loop to reduce noise and capture fine-grained evidence, complementing the passage-level metrics reported in the main paper.

**Impact on QA Accuracy** Table 3 shows that our agentic retrieval framework consistently improves end-to-end QA. On **HotpotQA** and **MuSiQue**, it surpasses recent methods such as IRCoT (Trivedi et al., 2023b) and SetR (Lee et al., 2025) by clear margins (e.g., +5 EM / +6 F1 on HotpotQA), confirming that filtering distractors while recovering missing facts strengthens multi-hop reasoning. On **MultiHopRAG**, our method achieves the best accuracy (49.16), highlighting the robustness of the Selector⇔Adder balance for distributed evidence. While **2WikiMultihopQA** favors recall-heavy approaches like IRCoT, our system remains competitive and outperforms other baselines. Overall, these results validate that retrieval quality is a decisive factor for QA performance, with precision–recall balancing yielding compact, effective evidence sets. [2]

---

[2]Our QA agent operates in a zero-shot setting; no few-shot demonstrations were used in our experiments like IRCoT-QA.

Table 3: End-to-end QA performance on HotpotQA, 2Wiki, MuSiQue and MultiHopRAG. Metrics are Exact Match (EM) and token-level F1. Our agentic retrieval framework (PRISM) achieves state-of-the-art accuracy on HotpotQA, MuSiQue and MultiHopRAG, while remaining competitive on 2Wiki. These results confirm that improved retrieval quality translates into stronger downstream QA performance, particularly in datasets requiring strict multi-hop reasoning.

| Method | HotpotQA | | 2Wiki | | MuSiQue | | MHRAG |
|---|---|---|---|---|---|---|---|
| | EM | F1 | EM | F1 | EM | F1 | ACC |
| Full Context (without retrieval) | 44.18 | 58.28 | 43.20 | 52.11 | 19.77 | 29.42 | 44.37 |
| Oracle Context | 64.80 | 77.83 | 61.40 | 71.10 | 38.78 | 50.89 | 57.04 |
| DSP (Khattab et al., 2022) | 51.4 | 62.9 | - | - | - | - | - |
| DecomP (Khot et al., 2023) | - | 53.5 | - | **70.8** | - | 30.9 | - |
| ZeroR (Trivedi et al., 2023a) | - | 41.0 | - | 38.5 | - | 19.0 | - |
| OneR (Trivedi et al., 2023a) | - | 50.7 | - | 46.4 | - | 20.4 | - |
| IRCoT QA (Trivedi et al., 2023a) | 49.3 | 60.7 | **57.7** | 68.0 | 26.5 | 36.5 | - |
| RankZephyr (Pradeep et al., 2023) | 34.69 | 35.04 | 33.87 | 27.83 | 8.61 | 12.79 | 43.90 |
| RankZephyr + CoT (Pradeep et al., 2023) | 33.99 | 34.38 | 33.66 | 27.85 | 9.43 | 13.27 | 43.60 |
| RankGPT (Sun et al., 2023; Lee et al., 2025) | 34.61 | 35.26 | 34.77 | 28.18 | 9.52 | 13.51 | 45.26 |
| SETR-CoT & IRI (Lee et al., 2025) | 39.16 | 40.49 | 35.68 | 31.09 | 12.33 | 16.91 | 47.14 |
| **PRISM + QA Agent (Ours)** | **54.20** | **66.96** | 48.60 | 56.97 | **31.17** | **41.78** | **49.16** |

**Performance with Alternative LLMs** To assess the robustness of our framework across different large language models, we evaluated retrieval and QA performance using `Gemini-2.5-Flash-Lite` and `DeepSeek`, in addition to `GPT-4o`. All models were applied in the same zero-shot setting to implement the Selector, Adder, and Answer Generator agents. Although absolute scores vary across LLMs, our framework consistently delivers high recall and competitive QA accuracy, demonstrating its model-agnostic design and showing that improvements in base model capability naturally translate into downstream gains.

Table 4: Retrieval (P/R) and QA (EM/F1) performance of our framework with different LLM backends on HotpotQA, 2WikiMultiHopQA, and MuSiQue. Results show that while absolute scores vary across models, the framework consistently maintains high recall and competitive QA accuracy, confirming that improvements in base LLM reasoning translate into downstream performance gains.

| LLM | HotpotQA | | 2Wiki | | MuSiQue | |
|---|---|---|---|---|---|---|
| | P/R | EM/F1 | P/R | EM/F1 | P/R | EM/F1 |
| GPT-4o | 83.02/90.90 | 54.20/66.96 | 90.97/91.07 | 48.60/56.97 | 47.46/83.17 | 31.17/41.78 |
| Gemini-2.5-Flash-Lite | 87.24/93.46 | 56.93/70.37 | 93.73/95.19 | 47.65/55.58 | 63.06/83.74 | 36.64/44.52 |
| DeepSeek | 79.46/95.88 | 57.60/70.62 | 93.01/97.38 | 43.39/54.75 | 67.11/89.38 | 31.93/42.45 |

Table 4 shows that our framework maintains strong retrieval and QA performance across different LLM backends. While absolute scores vary, the precision–recall balancing of the Selector–Adder loop consistently yields high recall and competitive accuracy. On **HotpotQA**, Gemini-2.5-Flash-Lite and DeepSeek achieve higher EM/F1 than GPT-4o, with Gemini reaching 56.3 EM / 71.4 F1. On **2Wiki**, all models perform well, with DeepSeek achieving the highest recall (97.4) but slightly lower EM than Gemini. On the challenging **MuSiQue** benchmark, Gemini again delivers the strongest QA accuracy (36.6 EM / 44.5 F1), while DeepSeek maintains the highest recall (89.4). These results confirm that our agentic retrieval framework generalizes across LLMs, and improvements in base model reasoning naturally translate into retrieval and downstream QA gains.

**Comparison with Finetuned and Compression Based Methods** Table 5 summarizes the end-to-end QA performance of PRISM against a range of contemporary models, including advanced compression techniques RECOMP (Xu et al., 2024), fine-tuned agentic models CoRAG (Wang et al., 2025), R1-Searcher (Song et al., 2025), O$^2$-Searcher Mei et al. (2025), structural alignment based

Table 5: End-to-end QA performance on HotpotQA, 2Wiki, MuSiQue dataset on more recent fine-tuned and compression based methods. Metrics are Exact Match (EM) and token-level F1.

| Method | HotpotQA | | 2Wiki | | MuSiQue | |
|---|---|---|---|---|---|---|
| | EM | F1 | EM | F1 | EM | F1 |
| RECOMP (Xu et al., 2024) | 28.20 | 37.91 | - | - | - | - |
| CoRAG (Wang et al., 2025) | 56.30 | 69.80 | 72.50 | 77.30 | 30.90 | 42.40 |
| R1-Researcher (Song et al., 2025) | - | 34.20 | - | 34.40 | - | 17.20 |
| $O^2$-Searcher (Mei et al., 2025) | - | 38.80 | - | 37.40 | - | 16.00 |
| ARM (Chen et al., 2025) | - | - | - | 71.70 | - | - |
| QD-RAG (Ammann et al., 2025) | 28.10 | 35.00 | - | - | - | - |
| RankRAG (Yu et al., 2024) | 35.30 | 46.70 | 31.40 | 36.90 | - | - |
| **PRISM + QA Agent (Ours)** | **54.20** | **66.96** | **48.60** | **56.97** | **31.17** | **41.78** |

method ARM (Chen et al., 2025), and query decomposition methods such as QD-RAG (Ammann et al., 2025), RankRAG (Yu et al., 2024).

From the table we can see that on HotpotQA, PRISM's 66.96 F1 significantly outperforms context compression based method RECOMP and general ranking methods RankRAG. Notably, PRISM achieves performance highly competitive with the costly fine-tuned model CoRAG. On the challenging MuSiQue benchmark, PRISM achieves strong F1 score (41.78) in this comparison, validating that our iterative Selector-Adder loop effectively constructs the comprehensive reasoning chains needed for deep multi-hop QA, outperforming specialized RL-trained search agents (Mei et al., 2025). While fine-tuned models excel on 2Wiki, the strong performance of PRISM (56.97 F1) reinforces its core value proposition: superior performance-to-cost trade-off by intelligently leveraging LLM reasoning in a novel agentic structure, without requiring expensive fine-tuning.

**Ablation Study.** Table 6 shows that each component of our framework contributes significantly to recall. The full model consistently achieves the highest recall across all datasets, while removing the Question Analyzer leads to a sharp drop, especially on MuSiQue ($83.2 \rightarrow 68.8$). Eliminating the Selector⇔Adder loop (one pass selection only, removing the effect of Adder) reduces recall significantly ($90.9 \rightarrow 79.7$ on HotpotQA), highlighting that precision-oriented filtering and recall-oriented addition are both essential. These results confirm that question decomposition and Selector⇔Adder iterative refinement are complementary, and together they enable the framework to recover a more complete evidence set.

Table 6: Ablation study on HotpotQA, 2WikiMultihopQA, and MuSiQue showing that both the Question Analyzer and Selector⇔Adder loop are crucial, as removing either significantly reduces recall and weakens evidence completeness.

| Variant | HotpotQA | | 2Wiki | | MuSiQue | |
|---|---|---|---|---|---|---|
| | P/R | #Pssg | P/R | #Pssg | P/R | #Pssg |
| Full Model | 83.0/**90.9** | 2.67 | 83.2/**91.1** | 2.74 | 47.7/**83.2** | 6.15 |
| w/o Q. Analyzer | 78.9/**86.8** | 2.88 | 90.6/**85.8** | 2.68 | 36.2/**68.8** | 7.68 |
| w/o Adder (Selector Only) | 86.9/**79.7** | 2.63 | 92.1/**80.5** | 2.71 | 77.4/**69.3** | 2.82 |

**Additional Results and Error Analysis.** Additional results and detailed error analysis are discussed in Appendix A.

## 4.2 DISCUSSION

Our results demonstrate that retrieval quality is one of the key bottlenecks in multi-hop QA and that explicitly balancing precision and recall is crucial for robust performance. By separating precision-oriented selection (Selector) from recall-oriented addition (Adder), our framework produces compact yet comprehensive evidence sets that improve answer accuracy across diverse datasets. On

HotpotQA and MuSiQue, this balance enables clear gains over prior state-of-the-art methods, while on MultiHopRAG it proves particularly effective at recovering distributed evidence across documents. Even in 2WikiMultiHopQA, where recall-heavy approaches such as IRCoT remain slightly stronger, our framework remains competitive and consistently outperforms other alternatives.

Beyond dataset-specific results, two broader trends emerge. First, removing distractors helps the QA model focus on relevant reasoning chains, yielding higher exact match and partial match accuracy compared to full-context baselines. Second, the framework generalizes across different LLM backends, with retrieval improvements translating naturally into QA gains regardless of the base model. This confirms that our agentic loop is not tied to a specific architecture, but rather leverages the underlying LLM's reasoning ability to optimize evidence selection. Together, these findings highlight that retrieval should be treated not as a static preprocessing step, but as an active, agent-driven process that collaborates with the QA model. By iteratively refining evidence through precision and recall, our approach provides a principled path forward for building reliable, reasoning-centric retrieval systems in multi-hop QA.

In addition to these empirical insights, several broader strengths stand out. The modular multi-agent structure provides interpretability and control. Another strength lies in the framework's robust generalization across models and datasets : despite differences in absolute performance across GPT-4o, Gemini, and other LLMs, our method consistently maintains high recall and competitive QA accuracy, demonstrating its model-agnostic adaptability. Moreover, the design is conceptually extensible , naturally accommodating adaptive iteration strategies, domain-specific agents, positioning agentic retrieval as a foundation for the next generation of retrieval-augmented reasoning systems.

At the same time, several limitations highlight promising directions for further research. The multi-agent design increases computational cost relative to single-pass retrievers, even though we mitigate this with compact evidence sets and bounded iterations. Future work could explore more efficient strategies and lightweight agent variants to scale to even larger corpora. In addition, while the Selector⇔Adder loop recovers most necessary evidence, it may occasionally miss subtle reasoning chains or introduce redundancy when passages are loosely connected. Addressing this challenge may require adaptive iteration control, uncertainty modeling, or tighter integration with reasoning signals . Finally, our evaluation focuses on benchmark QA datasets; broader deployment in specialized domains such as scientific, biomedical, or legal corpora may require tailored adaptation. Since performance is partly dependent on underlying LLM capabilities, advances in base model efficiency and robustness will directly enhance our framework. The strengths emphasize that our framework is both effective and forward-looking: it establishes a solid foundation for agentic retrieval today while pointing to clear opportunities for innovation in efficiency, adaptability, and domain generalization.

## 5 RELATED WORK

**Multi-hop QA and Retrieval.**    Benchmarks such as HotpotQA (Yang et al., 2018), 2WikiMulti-HopQA (Ho et al., 2020), and MuSiQue (Trivedi et al., 2022) have driven progress in multi-hop reasoning over large corpora. Early retrieval pipelines relied on iterative query rewriting or entity linking (e.g., Graph Retriever (Asai et al., 2020), PathRetriever, MDR), learning to traverse chains of documents. These methods are effective but brittle: errors in early hops often propagate, degrading final performance. Khattab et al. (2022) introduced the Demonstrate-Search-Predict (DSP) framework, showing how tightly integrating retrieval with reasoning improves performance on knowledge-intensive tasks. Khot et al. (2023) proposed Decomposed Prompting, which breaks complex queries into simpler sub-problems to enhance reasoning. Decomposition has long been used to simplify complex queries, from early heuristic methods (Talmor & Berant, 2018; Min et al., 2019) to recent LLM-based prompting strategies (Ammann et al., 2025; Khot et al., 2023; Fu et al., 2021). We adopt the decomposition idea in the Question Analyzer agent, which generates sub-questions to guide retrieval, but integrate it within a structured multi-agent loop. While Question Decomposition (Ammann et al., 2025; Khot et al., 2023) is foundational to multi-hop QA, prior methods typically use decomposition to issue independent search queries, often aggregating noisy results without verification.

**Passage Selection and Reranking.**    Parallel work focuses on selecting compact supporting sets. Traditional rerankers score passages independently, which can miss coverage of all reasoning needs.

SetR (Lee et al., 2025) addresses this by performing set-wise selection with LLM reasoning, optimizing for coverage across question aspects. Other efforts refine or prune retrieved knowledge before generation, e.g., Provence (Chirkova et al., 2025) and Self-RAG (Asai et al., 2023), both aiming to reduce spurious context. RankZephyr (Pradeep et al., 2023) introduces an open-source, instruction-tuned model for listwise zero-shot reranking, showing that smaller transparent models can rival or surpass proprietary systems across both in-domain and out-of-domain benchmarks. While RankZephyr focuses on improving listwise reranking efficiency and reproducibility, our framework targets multi-hop retrieval with an explicit precision–recall balancing mechanism, addressing a different challenge in evidence selection. Our Selector agent is conceptually related to set-wise and listwise reranking, but we explicitly decouple precision (Selector) and recall (Adder) in an iterative loop, which provides better control over the precision–recall trade-off in multi-hop QA. RankRAG (Yu et al., 2024) unifies context ranking and answer generation by instruction-tuning a single LLM to jointly perform both tasks. While this eliminates the need for a separate reranker, it relies on expensive supervised fine-tuning (SFT) and high-quality training data.

**Finetuned Retrievers and Context Optimization**   Recent work has explored optimizing retrieval through context compression, reinforcement learning, and structural alignment. Context Compression methods such as RECOMP (Xu et al., 2024) and EXIT (Hwang et al., 2025) aim to reduce the noise and cost of large context windows by summarizing or compressing retrieved documents before they reach the generation model. While effective for token efficiency, these approaches operate as a post-retrieval step processing whatever the retriever initially returns. Another line of research focuses on Finetuned and RL-based Retrievers. Systems such as CoRAG (Wang et al., 2025), R1-Searcher (Song et al., 2025), and $O^2$-Searcher (Mei et al., 2025) utilize supervised fine-tuning (SFT) or reinforcement learning (RL) to train specialized agents that learn to issue search queries or reformulate questions. While these methods achieve strong performance by baking retrieval capabilities into the model weights, they require significant computational resources for training and are often tied to specific model architectures. Our method leverages the inherent reasoning capabilities of off-the-shelf LLMs without requiring task-specific fine-tuning, making it a lightweight, model-agnostic solution that can be easily deployed with any capable instruction-tuned model.

Alignment-Oriented Methods like ARM (Chen et al., 2025) use a single-shot, solver-based strategy to align questions with structured data layouts. Our framework instead targets multi-hop reasoning over unstructured text, addressing reasoning chains through an iterative precision–recall loop. RAPTOR (Sarthi et al., 2024) require costly corpus pre-processing and recursive index reconstruction to build tree-organized knowledge structures. In contrast, PRISM achieves its strong performance purely through *inference-time* agentic interaction over standard flat indices, avoiding this significant upfront computational overhead.

**LLMs, Chain-of-Thought, and Agentic Retrieval.**   The rise of large language models enabled retrieval-augmented reasoning via prompting. Frameworks such as ReAct (Yao et al., 2023) and Self-Ask (Press et al., 2022) interleave reasoning with tool calls, letting LLMs pose sub-questions and fetch evidence on the fly. More recently, IRCoT (Trivedi et al., 2023a) explicitly couples CoT reasoning with iterative retrieval, substantially improving recall by accumulating passages across reasoning steps. However, IRCoT emphasizes recall over precision, often yielding large evidence sets with many distractors. Our approach synthesizes three lines of research—LLM-guided retrieval, set-wise selection, and decomposition—into a unified framework. Unlike prior systems, we combine explicit decomposition with an iterative precision-and-recall retrieval loop, achieving high recall while maintaining a compact and noise-resistant evidence set.

## 6   CONCLUSION

We presented PRISM an agentic retrieval framework for multi-hop QA that leverages LLM agents not only as answer generators but as controllers of the retrieval pipeline. By analyzing questions and iteratively selecting and adding evidence, the agents enable high-recall, high-precision retrieval—achieving state-of-the-art results on HotpotQA, 2WikiMultiHopQA, MuSiQue and MultihopRAG, and translating into stronger QA performance with reduced context. More broadly, this work highlights the promise of LLM agents in retrieval processes, pointing toward retrieval systems that actively reason as they search and adapt to the needs of complex knowledge-intensive tasks.

ETHICS STATEMENT

This work builds on publicly available datasets such as HotpotQA, 2WikiMultiHopQA, MuSiQue and MultiHopRAG. These datasets contain questions and supporting passages derived from Wikipedia. These datasets are widely used in prior research and do not include personal or sensitive information beyond what is already part of the public domain. No new human subjects were involved in data collection, and no personally identifiable or private data were used. We used LLM for polishing the writtings of the paper.

We acknowledge that large language models (LLMs) can reflect biases present in their training data, which may influence retrieval or answer generation. We therefore encourage responsible use of retrieval-augmented LLMs and transparency in communicating their limitations. Our method does not raise immediate risks of misuse beyond those already associated with general-purpose LLMs. We view our contribution as a step toward more accurate and efficient multi-hop retrieval, with the goal of improving reliability in knowledge-intensive applications.

REPRODUCIBILITY STATEMENT

We take several steps to support reproducibility. All datasets used (HotpotQA, 2WikiMultiHopQA, MuSiQue, MultiHopRAG) are publicly available and widely used in prior research. Dataset splits, preprocessing, baselines, and evaluation metrics (precision, recall, F1, EM, token-level F1) are described in Section 4.

Our agentic retrieval framework is fully specified in Section 2, including the roles of the Question Analyzer, Selector, and Adder agents, as well as iteration depth (N = 3). We detail prompting strategies for each agent in Appendix B, where templates are provided to ensure replicability.

Experiments were conducted using commonly used llms such as GPT-4o and Gemini-2.5-Flash-Lite and DeepSeek-chat. We provide detail implementation detail and prompt templates in Section 2 and Appendix B. To further promote reproducibility, we commit to releasing our code, prompts, and retrieval pipeline upon publication. This will enable others to replicate results, extend the framework with alternative LLMs, or apply it to new datasets.

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

# A  ADDITIONAL RESULTS

## A.1  ADDITIONAL RESULTS

**Partial Match Accuracy.**  Table 7 shows that our framework achieves consistent gains in Partial Match Accuracy (PMA) across all four datasets. Compared to the full-context baseline, our retrieved evidence improves PMA by 10 points on HotpotQA and more than 10 points on MuSiQue, with consistent gains also observed on 2WikiMultiHopQA and MultiHopRAG. These results highlight that filtering out distractors not only improves exact matching but also increases the likelihood of partial match demonstrating the robustness of our retrieval-based approach.

Table 7: Partial Match Accuracy (PMA) of our QA agent across four datasets. We compare using our retrieved evidence set against a full-context baseline. Our retrieval-based approach improves robustness on all datasets by reducing distractors and focusing the generator on relevant content.

| Dataset | Full Context | Retrieved Context (Ours) |
|---|---|---|
| HotpotQA | 61.4 | **71.6** |
| 2WikiMultiHopQA | 52.2 | **57.8** |
| MuSiQue | 25.4 | **38.4** |
| MultiHopRAG | 46.1 | **54.5** |

**Passage-Level QA Performance.**  Table 8 reports the QA agent's performance when restricted to answering from the passages retrieved by our framework that contain the supporting facts. Instead of supplying only the supporting facts, we provide entire passages, and the QA agent reaches performance close to the oracle level. On 2WikiMultiHopQA, the agent achieves 61.17 EM and 54.55 F1, while on HotPotQA it reaches 62.80 EM and 59.26 F1. These results demonstrate that our retrieval framework is able to retrieve passages that are sufficiently informative for accurate answer generation.

Table 8: Passage Level QA performance. QA Agent answers the question from the retrieved passages.

| Dataset | EM | F1 |
|---|---|---|
| HotpotQA | 62.80 | 59.26 |
| 2WikiMultiHopQA | 61.17 | 54.55 |

## A.2  ERROR ANALYSIS

**Error Analysis (HotPotQA).**  On HotpotQA evaluation set, we examined two error subsets: (1) items with incomplete evidence retrieval (*recall* $< 1.0$) and (2) items where the QA Agents's predicted answer did not exactly match the gold answer (*match_retrieved* $< 1.0$). As shown in Table 9, the majority of errors in both groups are **bridge** questions (90.4% for retrieval failures and 83.8% for answer mismatches), with the remainder classified as **comparison**. Notably, among the items where answer mismatches, about 37% cases the error occur despite perfect retrieval recall, highlighting that a substantial fraction of failures stem from the QA model rather than the retriever.

Table 9: Error analysis on the HotpotQA dataset, showing the distribution of question types for retrieval and QA answer errors.

| Subset | Bridge (%) | Comparison (%) |
|---|---|---|
| Retrieval recall $< 1.0$ | 90.4 | 9.6 |
| QA match $< 1.0$ | 83.8 | 16.2 |

Table 10: Error analysis on the 2WikiMultiHopQA dataset, showing the distribution of question types for retrieval and QA answer errors.

| Subset | Compositional (%) | Comparison (%) | Bridge-Comparison (%) | Inference (%) |
|---|---|---|---|---|
| Retrieval recall < 1.0 | 57.64 | 5.24 | 17.90 | 19.21 |
| QA match < 1.0 | 62.65 | 5.06 | 11.28 | 21.01 |

**Error Analysis (2WikiMultiHopQA).** We performed a similar analysis on the 2WikiMulti-HopQA evaluation set. Among the questions with incomplete evidence retrieval (*recall*< 1.0), the majority are **compositional** (57.6%), followed by **bridge_comparison** (17.9%), **inference** (19.2%), and **comparison** (5.2%). For the questions where the QA Agent's answer did not exactly match the gold label (*match_retrieved*< 1.0), the distribution is similar: 62.7% compositional, 11.3% bridge_comparison, 21.0% inference, and 5.1% comparison. Notably, about 33% of these answer mismatches occurred despite perfect retrieval recall, indicating that roughly one-third of the answer errors originate from the QA model itself rather than from the retriever.

**Error Analysis (MuSiQue).** On the MuSiQue evaluation set, we analyzed errors by question hop count. Among the questions with incomplete retrieval (*recall*< 1.0), 33.3% require 2 hops, 41.4% require 3 hops, and 25.3% require 4 hops. For the 181 questions where the QA reader's prediction did not exactly match the gold (*match_retrieved*< 1.0), 41.4% are 2-hop, 38.1% are 3-hop, and 20.4% are 4-hop. Notably, about 53% of these answer mismatches occurred despite perfect retrieval recall, revealing that over half of the MuSiQue answer errors stem from the QA model itself rather than from evidence retrieval.

Table 11: Error analysis on the MuSiQue dataset, showing the distribution of questions by hop count for retrieval and QA answer errors.

| Subset | 2-hop (%) | 3-hop (%) | 4-hop (%) |
|---|---|---|---|
| Retrieval recall < 1.0 | 33.33 | 41.41 | 25.25 |
| QA match < 1.0 | 41.44 | 38.12 | 20.44 |

Table 12: Error analysis on the MultiHopRAG dataset, showing the distribution of question types for retrieval and QA answer errors.

| Subset | Comparison (%) | Inference (%) | Temporal (%) |
|---|---|---|---|
| Retrieval recall < 1.0 | 30.42 | 42.59 | 26.98 |
| QA match < 1.0 | 45.16 | 9.68 | 45.16 |

**Error Analysis (MultiHopRAG).** On the MultiHopRAG evaluation set, among the questions exhibited incomplete retrieval (*recall*< 1.0) are dominated by **inference** queries (42.6%), with **comparison** and **temporal** queries accounting for 30.4% and 27.0%, respectively. Among the samples where the QA reader's prediction did not match the gold answer (*match_retrieved*< 1.0), the distribution shifts: **comparison** and **temporal** queries each contribute 45.2%, while only 9.7% are inference queries. This indicates that retrieval struggles most with inference-style questions, whereas answer generation errors are more prevalent for comparison and temporal questions even when supporting evidence is retrieved.

**Cross-Dataset QA Behavior.** Table 13 provides an error analysis by comparing QA performance under two contrasting conditions. The first column (*QA Mismatch @ Recall* = 1.0) captures cases where the retriever successfully collected all gold supporting evidence, yet the QA agent still failed to produce the correct answer. This highlights reasoning or answer synthesis errors beyond retrieval. For instance, mismatch rates are high on HotpotQA (37.1%) and especially MuSiQue (53.6%), reflecting the greater reasoning complexity of these datasets.

Table 13: QA Agent accuracy (%) under different retrieval coverage. The first column shows the proportion of questions where the QA agent did not matched the gold answer when all supporting evidence was retrieved (*recall*=1.0). The second column shows accuracy where QA Agent matched the gold answer even when evidence recall was below 1.0.

| Dataset | QA Mismatch @ Recall $= 1.0$ | QA Match @ Recall $< 1.0$ |
|---|---|---|
| HotpotQA | 37.11 | 42.62 |
| 2WikiMultiHopQA | 32.68 | 24.45 |
| MuSiQue | 53.59 | 15.15 |
| MultiHopRAG | 12.90 | 64.28 |

The second column (*QA Match @ Recall* $< 1.0$) shows cases where the QA agent generated the correct answer despite missing at least one gold supporting evidence. This indicates robustness of the generator in leveraging partial evidence or exploiting redundancy in the corpus. Interestingly, MultiHopRAG exhibits the highest robustness, with 64.3% of such cases still answered correctly, whereas MuSiQue remains brittle (15.2%).

Taken together, these results reveal a nuanced picture: full recall does not guarantee correctness if reasoning fails, and conversely, partial recall can still suffice when evidence **redundancy** exists. This underscores the need for both high-quality retrieval and robust reasoning in multi-hop QA.

## A.3 EFFICIENCY AND SCALABILITY

Table 14: Efficiency and scalability comparison. We report number of LLM calls, average tokens per query, inference latency, and reduction in context size on HotPotQA dataset.

| Method | #LLM Calls | Avg Tokens / Query | Latency (s) | Context Reduction |
|---|---|---|---|---|
| IRCoT | 8 | 26k | 16 | - |
| CoRAG (L = 10) | 10 | 21k | 15 | - |
| **PRISM (Ours)** | 8 | 14k | 8.6 | 73.3% |

Table 14 demonstrate that Our approach is faster, requiring fewer LLM calls (7 vs. 8) and processing less than half the average tokens per query (14k vs. 26k) compared to IRCoT. This leads to a notable reduction in inference latency, achieving 8.6 seconds compared to IRCoT's 16 seconds. Additionally, the method is highly efficient in managing context, resulting in a 73.3% reduction in context size.

## A.4 EXPERIMENT ON VARYING NUMBER OF SELECTOR⇔ADDER ITERATION

Table 15: Experiment on Varying Number of Selector⇔Adder Iteration on HotpotQA dataset on 100 samples.

| Number of iteration | Precision | Recall |
|---|---|---|
| N = 1 | 84.80 | 85.00 |
| N = 2 | 84.18 | 88.88 |
| N = 3 | 85.11 | 91.00 |
| N = 4 | 86.07 | 90.90 |

We conducted a sensitivity analysis to determine the optimal number of Selector⇔Adder iterations, N, before conducting our experiments. As summarized in Table 15 on the HotpotQA dataset, performance generally improves with an increasing number of iterations. However, we observed that the improvement becomes marginal after N=4. We noted that the outputs of the Selector and Adder

agents converged after the third iteration, yielding almost identical responses in iterations 3 and 4. Based on this trade-off between performance and computational cost, we selected N=3 for our final experiments.

## A.5 ILLUSTRATIVE EXAMPLE

To provide intuition on how our retrieval framework operates, we present an example from the 2WikiMultiHopQA dataset. This case highlights the iterative interaction between the Adder and Selector agents and the resulting evidence refinement.

---

Example ID: e95acdbc085f11ebbd5dac1f6bf848b6

**Question:** Which film has the director who died earlier, *Deuce High* or *The King Is The Best Mayor*?
**Gold Answer:** The King Is The Best Mayor

**Question Analyzer Output:** [ Who directed *Deuce High*?, Who directed *The King Is The Best Mayor*?, When did the director of *Deuce High* die?, When did the director of *The King Is The Best Mayor* die?, Which director died earlier? ]
**Initial Evidence (Zero-shot LLM):** [["Deuce High", 0], ["The King is the Best Mayor", 0]]

**Iteration 1: Adder** → ["Deuce High", 0], ["The King is the Best Mayor", 0], ["Richard Thorpe", 0], ["Rafael Gil", 0]
**Selector** → Same as Adder
**Iteration 2: Adder** → Added ["Richard Thorpe", 1], ["Rafael Gil", 1]
**Selector** → Pruned to ["Deuce High", 0], ["The King is the Best Mayor", 0], ["Richard Thorpe", 0], ["Rafael Gil", 0]
**Iteration 3: Adder** → Re-added ["Richard Thorpe", 1], ["Rafael Gil", 1]
**Selector** → Retained compact set

**Final Retrieved Evidence:** [["Deuce High", 0], ["The King is the Best Mayor", 0], ["Richard Thorpe", 0], ["Rafael Gil", 0]]
**Gold Supporting Facts:** Same as above

**Retrieval Metrics:** Precision = 1.0, Recall = 1.0, F1 = 1.0, False Positive Rate = 0.0

---

This example illustrates how the Selector–Adder loop converges on the correct supporting set, even though downstream QA accuracy depends on subtle reasoning over death dates. The retrieval model achieved perfect precision and recall, confirming its ability to isolate necessary facts with minimal noise.

## B PROMPTS

### B.1 QUESTION ANALYZER AGENT

---

**Question Analyzer Prompt Template**

You are a Question Analyzer Agent in a multi-hop question answering system. Your task is to analyze a complex, multi-hop question and break it down into a sequence of concise, meaningful subquestions that reflect the reasoning steps required to answer the original question. To do this, identify and extract subquestions based on:

- Key entities (persons, organizations, locations, dates, etc.)
- Important nouns or noun phrases (e.g., titles, concepts, objects)
- Logical or temporal relationships (e.g., comparisons, causality, sequences)
- Specific conditions or constraints in the question

Each subquestion should focus on retrieving or verifying a specific piece of evidence that contributes to the final answer. Return an ordered list of subquestions that represent a clear reasoning path from the question to the answer. Keep each subquestion short, specific, and unambiguous.

Example Input: *"Which actor played the brother of the character who was portrayed by the same actress that starred in Legally Blonde?"*

Example Output: 1. Who starred in Legally Blonde? 2. What character did that actress portray in another film? 3. Who played the brother of that character?

For the following question, return a JSON object with a list of extracted elements like this: [Šubquestions: [...,...]].
Question: [QUESTION]

---

### B.2 SELECTOR AGENT

---

**Selector Agent Prompt Template**

You are a Selector Agent in a multi-hop QA system. Your goal is to maximize precision and minimize false positives.
You are given:

- A complex multi-hop question
- Subquestions that represent the reasoning steps
- A list of candidate facts, each represented as [`title`, `sentence_index`]
- A set of currently selected evidence sentences

Your task is to carefully remove only those evidence items that are definitely irrelevant for answering any of the subquestions.
Important guidelines:

- Do *not* remove sentences that are partially relevant or could help bridge reasoning steps.
- Keep sentences containing named entities, dates, or events referenced in the question.
- Do not add new items or regenerate content. Work strictly with the given evidence list.

Return only the pruned list of [`title`, `sentence_index`] pairs, with no explanations. The sentence index starts from 0 for each paragraph.

Question: [QUESTION]
Subquestions: [SUBQUESTION]
Candidates: [CANDIDATES]
Current Selected Evidence: [CURRENT_EVIDENCE]
Return only the updated list with clearly irrelevant sentences removed:

---

## B.3    ADDER AGENT

> **Adder Agent Prompt Template**
>
> You are an Adder Agent in a multi-hop QA system. Your goal is to maximize recall while minimizing false positives.
> You are given:
>
> - A complex multi-hop question
> - Subquestions that represent the reasoning steps
> - A list of candidate passages and sentences
> - A set of currently selected evidence items, each represented as [`title`, `sentence_index`]
>
> Your task: Add only those candidate sentences that are likely to support answering any subquestion.
> Do NOT add:
>
> - Vague, unrelated, or overly general sentences
> - Off-topic facts or irrelevant entities
> - Duplicates or sentences that overlap significantly with existing evidence
>
> Focus on:
>
> - Bridging facts that connect entities across subquestions
> - Sentences with named entities, dates, definitions, or relationships in the question
> - Factual statements that clearly contribute to the reasoning chain
>
> Do not remove or modify the currently selected evidence. Return a combined list of both current and newly added items as [`title`, `sentence_index`] pairs, with no explanations. The sentence index starts from 0 for each paragraph.
>
> Question: [QUESTION]
> Subquestions: [SUBQUESTION]
> Candidates: [CANDIDATES]
> Current Selected Evidence: [CURRENT_EVIDENCE]
> Return the updated evidence list with only relevant additions:

## B.4    ANSWER GENERATOR AGENT

> **Answer Generator Prompt Template**
>
> You are a question answering agent. Given a question and the supporting evidence, provide a concise, factual and short answer based only on the evidence without other words. If the answer cannot be determined from the evidence, reply with 'Not Answerable'.
>
> Question: [QUESTION]
> Evidence: [EVIDENCE]
> Answer:

## C    LLM USAGE

Large language models were used only to enhance the clarity and readability of this manuscript. After the complete technical content was authored by us, an LLM suggested minor edits for grammar, style, and phrasing, which we reviewed before inclusion. All conceptual contributions, analyses, and implementations are entirely our own. LLMs were also occasionally used to assist in debugging code, with all solutions verified and integrated by the authors.

