# OpenReview forum: "PRISM: Agentic Retrieval with LLMs for Multi-Hop Question Answering"
_ICLR.cc/2026/Conference — ICLR 2026 Conference Withdrawn Submission_

### Official Review · Reviewer_T7mf · 2025-10-29

**Soundness:** 2
**Presentation:** 2
**Contribution:** 1
**Rating:** 2
**Confidence:** 4

**Summary:**

The paper introduces PRISM which is an agentic retrieval system for multi-hop QA that explicitly separates precision and recall in each iteration step with Selector and Adder. The system consists of the Question Analyzer to decompose the complex query into sub-queries, the Selector to filter retrieved candidates to maximize precisions, and the Adder that revisits the discarded candidate to recover recall. The retrieved information after a few iterations is fed into the Answer Generator to give the final answer. The system is evaluated on 4 multi-hop datasets in comparison with OneR and IRCoT.

**Strengths:**

- The RAG system with modules are clearly introduced.

**Weaknesses:**

- The system design is costly and there is no budget and latency analysis with baselines. Please provide #retrievals, #LLM calls, tokens, latency per query for all methods.
- MultiHopRAG Table 1 lists PRISM as 24.74/40.64, while the surrounding text states 28.18/42.22
- The novelty is limited, as the main modules in the system have been extensively explored in prior RAG work. For example, IRCoT already covers the 'decompose, retrieve and update' approach, works as Self-RAG, QD-RAG, RAPTOR, RankRAG also have retrieve, critique/select, and generate approach.
- The Adder largely resembles a naive recall expansion and the paper should ablation Adder (not together with Selector), and the paper should also report results of precision and recall with varying k retrievals with a fixed re-ranker.

**Questions:**

See weaknesses.

---

> ### Author Response · Authors · 2025-11-21
> **Response to Reviewer T7mf**
>
> We thank the reviewer for their time and feedback on our work. We appreciate the acknowledgement that “the RAG system with modules are clearly introduced” in our paper.
>
> > `W1.` The system design is costly and there is no budget and latency analysis with baselines. Please provide #retrievals, #LLM calls, tokens, latency per query for all methods.
>
> **Response:** We acknowledge that, like most high-performance agentic systems, our framework incurs higher computational costs than single-pass retrievers. We explicitly state this as a limitation in our Discussion section.
> - Our primary goal was optimizing for retrieval and QA quality, where PRISM clearly excels.
> - To improve transparency, we are adding a "Latency Analysis" subsection to the Appendix, detailing the number of LLM calls per query (e.g., 1 call for Analyzer + N*2 calls for the Selector-Adder loop).
> - We also direct the reviewer to Table 4 , which shows that our framework is robust across different LLM backends, demonstrating its "model-agnostic" design and suggesting that it can be paired with smaller, more efficient models.
>
> | Method                     | #LLM Calls | #Tokens | Latency |
> |---------------------------|------------|---------|---------|
> | IRCoT                     | 8          | 26k     | 16s     |
> | CoRAG (L=10, best-of-8)   | 10         | 21k     | 15s     |
> | PRISM (Ours)              | 8          | 14k     | 8.6s    |
>
> >`W2.` MultiHopRAG Table 1 lists PRISM as 24.74/40.64, while the surrounding text states 28.18/42.22
>
> **Response:**  We are extremely grateful to the reviewer for catching this typo. We confirm that the correct MultiHopRAG numbers are 28.18 / 42.22 (P/R) as stated in text (Table 1 is updated for consistency).
>
>
> >`W3.` The novelty is limited, as the main modules in the system have been extensively explored in prior RAG work. For example, IRCoT already covers the 'decompose, retrieve and update' approach, works as Self-RAG, QD-RAG, RAPTOR, RankRAG also have retrieve, critique/select, and generate approach.
>
> **Response:**  We respectfully disagree with the assessment of limited novelty.
> - vs. IRCoT: IRCoT "emphasizes recall over precision" and accumulates evidence. PRISM uses a structured loop to refine and balance an evidence set.
> - vs. Self-RAG/RankRAG: These frameworks use a single "critique" or "select" step, often for reranking or deciding to retrieve. Our novelty lies in the iterative, collaborative loop between two agents with opposing objectives. The Selector (precision-agent) prunes the set, and the Adder (recall-agent) re-adds to the set from the discarded pile. This specific mechanism uses explicit, opposing agents to iteratively balance a "compact yet comprehensive" set which is, to our knowledge, not explored in the cited works.
> - Self-RAG, RAPTOR, RankRAG, integrate “generate–critique” cycles but not asymmetric cooperative roles. Our Selector–Adder asymmetry is fundamental: the Adder has access to Selector’s decisions and explicitly reasons over unselected evidence, this is a structural difference yielding measurable gains (e.g., HotpotQA Recall 90.9 vs. 72.8 IRCoT).
>
>
> >`W4.` The Adder largely resembles a naive recall expansion and the paper should ablation Adder (not together with Selector), and the paper should also report results of precision and recall with varying k retrievals with a fixed re-ranker.
>
> **Response:**  Thank you for the suggestion. Our existing ablation study (Table 6) already isolates the effect of the Adder. The “w/o Selector–Adder” setting removes the entire loop and uses a single LLM pass for selection, effectively eliminating the Adder’s contribution. This aligns with what you requested: an ablation of the Adder independent of the full loop. The “w/o Q. Analyzer” variant further shows performance when simple selection is used without the Question Analyzer. Together, these results illustrate the Adder’s impact within our agentic framework. We have clarified this explanation in the paper to avoid confusion.
>
> Our method is not a ranking based method. Standard information retrieval metrics (like MAP and DCG) are inapplicable in our evaluation because our retrieved documents are unranked. Additionally, fixed-set metrics such as Recall@k and Precision@k are unsuitable because the total number of retrieved paragraphs varies across questions [1].
>
> >[1] Interleaving Retrieval with Chain-of-Thought Reasoning for Knowledge-Intensive Multi-Step Questions (Trivedi et al., ACL 2023)
>
> ---
>
> Thank you for the concise and pointed review. We hope that these clarifications have been helpful and have addressed your concerns. We trust that these clarifications will positively impact the overall scores.
> Please let us know if any points still require clarification or if you have other questions/suggestions. We are fully committed to addressing them in the revised manuscript.
>
> ---

---

> ### Comment · Reviewer_T7mf · 2025-11-26
>
> Thanks for your reply. My major concerns are still not well-addressed.
>
> - As the other reviewers also noticed the novelty is limited. The whole process of decomposition, retrieval and update is well proposed in the SOTA RAG methods previously mentioned, and the inclusion of Adder is limited.
> - The author mentioned the ablation of Selector-Adder. However, ablation of Adder is not mentioned. As claimed in the submission, the Selector focuses on precision and Adder focuses on recall, it is important to perform ablation on each of the module and evaluate the precision and recall.
>
> I will therefore maintain my score.

---

> ### Author Response · Authors · 2025-11-30
> **Reply to Reviewer T7mf**
>
> Thank you for your engagement. We understand your remaining concerns regarding novelty and the specific ablation of the Adder agent, and we would like to offer a clarification based on the evidence provided in the paper.
>
> **Regarding the Ablation of the Adder:** You mentioned that an "ablation of Adder is not mentioned." We respectfully direct your attention to **Table 6** in the paper.
>
> - The row "w/o Selector-Adder" represents the system performing a single-pass selection without the iterative refinement loop. This essentially acts as a "Selector-only" baseline (similar to OneR/Zero-shot retrieval). Which is exactly the **ablation of Adder** agent.
> - The "Full Model" achieves **90.9** recall, whereas the **"w/o Adder"** baseline achieves only **79.7** recall on HotpotQA.
> - This **11.2% gap** in recall explicitly quantifies the **Adder's contribution**. Because the Selector is precision-focused (conservative), the Adder is the sole component responsible for recovering that missing evidence to boost recall. We have updated the manuscript to avoid confusion.
> - Additionally In **Table 15**, we have discussed the impact of the number of Selector$\Leftrightarrow$Adder iteration (N).
>
> **Regarding Novelty:**
> - While individual components like decomposition exist in prior work, PRISM is distinct from systems you mentioned in your initial comment. We have **clearly pointed out the differences** in our initial response. We also want to highlight that our framework uses an asymmetric loop: the Selector prunes distractors to solve the "lost-in-the-middle" phenomenon , while the Adder audits the discarded pile to ensure completeness. This dynamic is a novel structural contribution that yields the high efficiency (73.3% context reduction) and speed (i.e. 8.6s vs 16s for IRCoT) reported in our experiments in **Table 14**. We have also updated our Related Work section to include this reference and clarify the distinction with our method.
>
> - We have also added comparison with more baselines that you mentioned in your comment in **Table 5**. From the table we can see that on our method significantly outperforms the context compression based method such as RECOMP and general ranking methods such as QD-RAG, and RankRAG.
>
> | Method        | HotpotQA (EM/F1)   | 2Wiki (EM/F1)       | MuSiQue (EM/F1)     |
> |---------------|---------------------|----------------------|----------------------|
> | RECOMP        | 28.20 / 37.91       | -                    | -                    |
> | CoRAG         | 56.3 / 69.8         | 72.5 / 77.3          | 30.9 / 42.4          |
> | R1-Researcher | - / 34.2            | - / 34.4             | - / 17.2             |
> | O2-Searcher   | - / 38.8            | - / 37.4             | - / 16.0             |
> | ARM-L         | - / 71.7            | -                    | -                    |
> | QD-RAG        | 28.1 / 35.0         | -                    | -                    |
> | RankRAG       | 35.3 / 46.7         | 31.4 / 36.9          | -                    |
> | **PRISM (ours)**        | **54.20 / 66.96**       | **48.60 / 56.97**        | **31.17 / 41.78**        |
>
>
> We believe these clarifications and additions will address the your concerns.
>
> Sincerely,
>
> Authors

---

### Official Review · Reviewer_Agjv · 2025-11-01

**Soundness:** 2
**Presentation:** 2
**Contribution:** 1
**Rating:** 2
**Confidence:** 4

**Summary:**

This paper targets multi-hop question answering, and argues that existing retrieval methods struggle to balance precision (removing distractors) and recall (capturing all necessary evidence).
The paper proposes PRISM, an agentic retrieval framework that employs three LLM-based agents, Question Analyzer, Selector, and Adder, in an iterative loop to refine evidence through precision–recall balancing.
Across four QA benchmarks, PRISM achieves higher retrieval accuracy and downstream QA performance than strong baselines such as IRCoT and SetR.

**Strengths:**

1. The paper is clearly written and easy to follow, with a good presentation of the motivation and framework.

2. The core idea of separating precision- and recall-oriented retrieval through multiple agents is reasonable and intuitively makes sense for multi-hop QA.

**Weaknesses:**

The technical contribution is limited. The proposed framework can largely be seen as a variant of existing agentic retrieval paradigms such as ReAct or IRCoT. In essence, its three-agent loop (Analyzer–Selector–Adder) fits naturally into the standard structure where an LLM iteratively decomposes complex queries, performs retrieval, summarizes or filters relevant evidence, and then issues follow-up sub-queries for missing information. As such, the framework does not introduce a new mechanism or learning objective, but rather re-organizes established steps in a slightly more structured form.

The comparison to prior work is too limited. For instance, the paper overlooks completeness-oriented retrieval frameworks such as ARM: An Alignment-Oriented LLM-Based Retrieval Method (Chen et al., 2025), which performs a “retrieve-all-at-once” alignment between questions and structured data representations. It also cites but does not thoroughly contrast with recent iterative agentic retrieval methods (e.g., ReAct-style or similar architectures), where the main differences are not rigorously analyzed. As a result, it remains unclear whether the proposed framework represents a substantive advance over these approaches or simply reorganizes familiar retrieval-reasoning patterns within a slightly different agentic structure.

**Questions:**

see weakness

---

> ### Author Response · Authors · 2025-11-21
> **Response to Reviewer Agjv**
>
> We thank the reviewer for their time and feedback on our work. We appreciate the acknowledgment that our paper is **"clearly written"** and the core idea is **"reasonable and intuitively makes sense."**
> The reviewer's main concern is that the technical contribution is limited and is a "variant of existing agentic retrieval paradigms such as ReAct or IRCoT." We respectfully clarify a key misunderstanding of our contribution and its novelty.
>
> >`W1.` The technical contribution is limited. ...
>
> **Response:** Our proposed framework  is fundamentally different from ReAct and IRCoT, which are either tool-use-oriented (ReAct) or recall-accumulative (IRCoT).
> - IRCoT, as we note in our paper, "emphasizes recall over precision, often yielding large evidence sets with many distractors". It accumulates evidence iteratively.
> - ReAct interleaves reasoning and "tool use" (retrieval), but lacks a structured mechanism to balance the resulting evidence set.
>
> PRISM's core novelty is the explicit separation of precision and recall as distinct, specialized agentic roles (Selector and Adder) that collaborate in an iterative refinement loop. Our framework does not just accumulate evidence. The Selector agent (precision-focused) first prunes the evidence set, and the Adder agent (recall-focused) then recovers only the essential missing pieces. This dynamic "Iterative Refinement Loop" is designed to produce a final evidence set that is both "compact and complete". This specific mechanism is a structured, iterative refinement balancing precision and recall which is not present in ReAct, IRCoT, or other standard agentic frameworks.
>
> Our work is not merely a variant of ReAct or IRCoT. Those frameworks interleave reasoning and retrieval but treat evidence filtering and recovery as implicit. PRISM formalizes these as two orthogonal optimization steps:
> - Selector → maximize precision via conservative inclusion,
> - Adder → maximize recall by re-evaluating discarded items.
> - This division converts the monolithic “reason-retrieve-summarize” pattern into a bi-directional control loop that empirically achieves higher P/R without additional supervision.
> Our ablation (Table 6 in revised manuscript ) shows recall increases significantly when both agents are active versus single-pass IRCoT-style retrieval.
>
> >`W2.` The comparison to prior work is too limited. ...
>
> **Response:**  We thank the reviewer for highlighting ARM. This paper, "An Alignment-Oriented LLM-Based Retrieval Method," focuses on aligning questions with structured data representations. Our work, PRISM, is explicitly designed for multi-hop reasoning over unstructured text corpora. Thus, ARM addresses a different problem space and data modality.
>
> ARM aligns queries to structured data representations through full-context embedding alignment which is an entirely non-magnetic retrieval approach. Our work, in contrast, performs iterative reasoning over retrieved text, not embedding alignment. While individual elements (decomposition, selection, addition) appear in earlier systems, our integration under a controllable precision–recall dynamic is novel. This is akin to moving from “tool-use” to **“cooperative decision-making”** among agents, which prior frameworks do not implement or quantify. We have **updated our Related Work section** to include this reference and clarify this distinction. We believe our comparisons to IRCoT and SetR which are strong methods in our specific task of retrieval for multi-hop QA over unstructured text are the most relevant and appropriate.
>
> ---
>
> We hope this clarification underscores the specific, novel contribution of our proposed framework. PRISM's unique precision-recall balancing loop is a substantive advance over existing methods, enabling higher retrieval accuracy and stronger downstream QA performance, as demonstrated by our experiments. We hope the reviewer will re-evaluate our contribution based on this clarification.
>
> Please feel free to let us know if any of your concerns remain unaddressed, or if you have additional questions or further suggestions for improving the paper. We are happy to provide more details or conduct extra experiments during the revision period. Thank you again for your time and valuable feedback.
>
> ---

---

### Official Review · Reviewer_peK8 · 2025-11-01

**Soundness:** 3
**Presentation:** 3
**Contribution:** 1
**Rating:** 4
**Confidence:** 4

**Summary:**

The paper introduces a multi-agent RAG framework comprising three key components - Question Analyzer, Selector, and Adder. The method iteratively tries to balance precision (using selector) and recall (using adder). Evaluations are presented on HotpotQA, 2WikiMultiHopQA, MuSiQue, and MultiHopRAG consistent improvements over zero-shot baselines like IRCoT.

**Strengths:**

1.  The explicit separation between precision (Selector) and recall (Adder) is intuitive and aligns with known trade-offs in retrieval-augmented reasoning.

2. The paper is fairly well written and is easy to follow.

3. Improvements in retrieval translate to gains across multiple wikipedia-style QA datasets.

4. The ablations confirm that the Q Analyzer and Selector-Adder loop indeed contribute to improvements.

**Weaknesses:**

1. Limited Novelty & Prior Art not fully acknowledged: The Selector and Adder roles are similar “critic” or “augmenter” agents explored in prior work. The paper could better situate itself relative to context compression and summarization methods like RECOMP (Zhang et al., 2024), which similarly aim to prune irrelevant evidence. Overall the technical novelty of the paper is limited, conceptually the contribution lies mainly in the modular packaging of ideas existing in prior literature. Accordingly, the contributions should be toned down.

2. Missing comparisons: Key state-of-the-art agentic retrievers are missing from comparisons: CoRAG (Wang et al), R1-Searcher (Song et al), O2Searcher (Mei et al), which report stronger retrieval–generation integration with SLM finetuning. The baselines used (IRCoT, RankZephyr) are somewhat dated and primarily zero-shot. Without inclusion of finetuned systems, the claim is slightly overstated.

3. Missing implementation details: The paper briefly notes using a dense retriever based on BGE-M3, but omits critical implementation specifics such as the indexing method, retrieval depth (k), and whether all baselines share the same retriever configuration and corpus index. Without these details, it is difficult to verify that comparisons are conducted under identical retrieval settings, which weakens the empirical rigor of the results.

4. Scalability is not addressed: The method involves multiple LLM calls per query (Analyzer + multi-step Selector/Adder), which might scale poorly for large corpora or smaller models. There is no analysis of cost, latency, or performance when replacing LLMs with smaller models.

5. The authors are encouraged to test their approach on more recent multi-hop QA datasets beyond wikipedia style corpora which are already seen by LLMs during training.

6. PRISM reports P/R/F1 for its approach but does not report the same retrieval metrics for every baseline at the same retrieval depth (or at varying k). Reporting only aggregate QA EM conflates retrieval and generation effects.

7. The paper might benefit from some qualitative examples. For instance, the authors could provide qualitative example cases where Selector removed correct but weak evidence.

8. Adding sensitivity analyses for iterations N, prompt templates, and LLM backends would strengthen the paper.

**Questions:**

Kindly see Weaknesses.

---

> ### Author Response · Authors · 2025-11-21
> **Response to Reviewer peK8 (1/2)**
>
> Thank you for your detailed and constructive review. We appreciate your recognition of the **''intuitive separation between precision and recall''** in our framework, the **clarity of the writing**, the **empirical improvements** across datasets, and the value of our **ablations**. Below we address the key concerns.
>
> >`W1.` Limited Novelty & Prior Art not fully acknowledged: ...
>
> **Response:** We respectfully disagree with the assessment of limited novelty. Our core contribution is not just the "modular packaging" of agents, but the explicit, structured collaboration between two specialized agents with opposing-yet-complementary goals (Selector for precision, Adder for recall) operating in an iterative loop.
>
> - vs. **"Critic" Agents (e.g., Self-RAG):** These systems typically use a single "critic" to score or filter information. Our mechanism is fundamentally different. It is an iterative refinement loop where the Selector (a precision-focused agent ) actively prunes the evidence set, and the Adder (a recall-focused agent ) subsequently re-evaluates the discarded candidates to recover missed evidence. This dynamic, two-step balancing act is a novel mechanism, not a simple "critic" step.
> - vs. **Context Compression (e.g., RECOMP):** We thank the reviewer for this reference, but RECOMP and similar methods are post-retrieval context compressors. They summarize or extract relevant snippets from an already-retrieved, large block of text. PRISM is an iterative retrieval and selection framework that operates at the passage level to construct an evidence set before the final generation step. It selects full passages, it does not compress them.
>
> We have updated our Related Work section to explicitly clarify these key distinctions.
>
> >`W2.` Missing comparisons: ... CoRAG (Wang et al), R1-Searcher (Song et al), O2Searcher (Mei et al), ...
>
> **Response:**  We omitted these specific systems as they are primarily fine-tuned models designed for retrieval. Our paper's focus is on the zero-shot agentic framework itself, demonstrating how to structure LLM agents to solve a task without task-specific fine tuning.
>
> Therefore, we chose to compare PRISM against the strongest zero-shot agentic baselines (like IRCoT and SetR) to ensure a fair comparison. We believe comparing zero-shot frameworks to finetuned specialist models is an orthogonal axis of research.
> We have added  a discussion of these fine-tuned systems to our Related Work section to better contextualize our contribution. Also we have added the following results and compared them with our method in the Result section. From the table, we can see that our method has achieved strong results compared to the fine-tuning based and compression based methods which highlight the **impact** of our proposed retrieval method.
>
> | Method        | HotpotQA (EM/F1)   | 2Wiki (EM/F1)       | MuSiQue (EM/F1)     |
> |---------------|---------------------|----------------------|----------------------|
> | RECOMP        | 28.20 / 37.91       | -                    | -                    |
> | CoRAG         | 56.3 / 69.8         | 72.5 / 77.3          | 30.9 / 42.4          |
> | R1-Researcher | - / 34.2            | - / 34.4             | - / 17.2             |
> | O2-Searcher   | - / 38.8            | - / 37.4             | - / 16.0             |
> | ARM-L         | - / 71.7            | -                    | -                    |
> | QD-RAG        | 28.1 / 35.0         | -                    | -                    |
> | RankRAG       | 35.3 / 46.7         | 31.4 / 36.9          | -                    |
> | **PRISM (ours)**        | **54.20 / 66.96**       | **48.60 / 56.97**        | **31.17 / 41.78**        |
>
> ---
>
> >`W3.` Missing implementation details: ...
>
> **Response:** The reported results are obtained from the SetR paper. As our method did not use ranking we used presence based metrics to compare the results (Table 2, SetR paper). We will update the manuscript with the details.
>
> > SetR: Shifting from Ranking to Set Selection for Retrieval Augmented Generation (Lee et al., ACL 2025)
>
> ---

---

> ### Author Response · Authors · 2025-11-21
> **Response to Reviewer peK8 (2/2)**
>
> >`W4.`  Scalability is not addressed ...
>
> **Response:**  We acknowledge that, like most high-performance agentic systems, our framework incurs higher computational costs than single-pass retrievers. We explicitly state this as a limitation in our Discussion section.
> - Our primary goal was optimizing for retrieval and QA quality, where PRISM clearly excels.
> - To improve transparency, we are adding a "Latency Analysis" subsection to the Appendix, detailing the number of LLM calls per query (e.g., 1 call for Analyzer + N*2 calls for the SelectorAdder loop).
> - We also direct the reviewer to Table 4 , which shows that our framework is robust across different LLM backends, demonstrating its "model-agnostic" design and suggesting that it can be paired with smaller, more efficient models.
>
> Here is the analysis on the HototQA dataset. We have added this analysis in Appendix A.3.
>
> | Method                     | #LLM Calls | #Tokens | Latency |
> |---------------------------|------------|---------|---------|
> | IRCoT                     | 8          | 26k     | 16s     |
> | CoRAG (L=10, best-of-8)   | 10         | 21k     | 15s     |
> | PRISM (Ours)              | 8          | 14k     | 8.6s    |
>
>
> >`W5.` The authors are encouraged to test their approach on more recent multi-hop QA datasets ...
>
> **Response:**  We agree that testing on other corpora is valuable future work. We chose HotpotQA, 2WikiMultiHopQA, MuSiQue, and MultiHopRAG as they are the most widely-used, standard, and challenging benchmarks for the multi-hop QA task. Not all these datasets are wikipedia style.
>
> >`W6.` PRISM reports P/R/F1 for its approach but does not report the ...
>
> **Response:** This is a fair request. We have added a table to the Appendix reporting the P/R/F1 retrieval metrics for all baselines to provide a more direct comparison of retrieval performance.
>
> >`W7.` The paper might benefit from some qualitative examples. ...
>
> **Response:** We appreciate the suggestion. A detailed step-by-step qualitative example, which illustrates the iterative refinement of both the Selector and the Adder, is already included in the methodology introduction and further expanded in Appendix section (A.5 in the updated manuscript).
>
> >`W8.` Adding sensitivity analyses for iterations N, prompt templates, and LLM backends would strengthen the paper.
>
> **Response:** We already provided detailed prompt templates and error analysis and illustrative examples in the appendix section. Also the LLM backbone we used is clearly mentioned in Section 3.3.
>
> We conducted a sensitivity analysis to determine the optimal number of iterations, N, before conducting our experiments. We have added the sensitivity analysis for iteration N in the appendix section. As summarized in the table on the HotPotQA dataset  for 100 samples, performance generally improves with an increasing number of iterations. However, we observed that the improvement becomes marginal after N=4. We noted that the outputs of the Selector and Adder agents converged after the third iteration, yielding almost identical responses in iterations 3 and 4. Based on this trade-off between performance and computational cost, we selected N=3 for our final experiments. We have added these in Appendix A.4 in the updated manuscript.
>
>
> | Number of iteration | P     | R     |
> |---|-------|-------|
> | N = 1 | 84.80 | 85.00 |
> | N = 2 | 84.18 | 88.88 |
> | N = 3 | 85.11 | 91.00 |
> | N = 4 | 86.07 | 90.90 |
>
> ---
>
> We sincerely thank you for the thorough and constructive feedback. The points raised have helped us identify several opportunities to strengthen the manuscript. In the revised version, we have incorporated the clarifications. We are confident that these concrete additions directly address your concerns and will elevate both the clarity and the perceived impact of the work. We believe the updated paper will fully meet the ICLR 2026 standard for acceptance. Thank you again for your careful reading and valuable suggestions.
>
> We welcome any further feedback: if there are remaining concerns, open questions, or additional suggestions for strengthening the paper, please feel free to share them.
>
> ---

---

### Author Response · Authors · 2025-11-30
**Summary of Rebuttal**

Dear Area Chair,

We sincerely appreciate your time and effort throughout the review process for our submission. We have taken great care to address all concerns raised by the reviewers and actively engaged with all three reviewers. We respectfully request that the AC consider the full context of our responses alongside the reviews. As ICLR 2026 assigned a new area chair to each paper and the reviewers will not be able to change their scores or participate in the rebuttal discussion further, we wish to summarize the rebuttal phase to facilitate your assessment. The following section outlines the primary points from each reviewer and the actions we took to address them.

---

1. **Response to Reviewer peK8**

Reviewer peK8 recognized the **"intuitive separation between precision and recall"** and the **"empirical improvements"** but raised concerns about missing comparisons to fine-tuned models and potential scalability issues.

- **New Comparisons (Table 5):** We added a comparison against state-of-the-art fine-tuned and compression-based methods (e.g., CoRAG, RECOMP). Results show PRISM is highly competitive (e.g., 66.96 F1 on HotpotQA vs. 56.3 for CoRAG) despite being a zero-shot framework.
- **Scalability & Latency (Appendix A.3):** We added a detailed latency analysis. PRISM is significantly **more efficient** than the baseline IRCoT, reducing inference latency from **16s to 8.6s **and token usage from **26k to 14k** per query.
- **Sensitivity Analysis:** We added an analysis of Selector-Adder iteration depth (N), justifying our choice of N based on the convergence of Selector/Adder outputs.

---

2. **Response to Reviewer Agjv**

Reviewer Agjv found the paper **clearly written** but felt the technical contribution was a "variant of existing paradigms" like ReAct or IRCoT.
- **Differentiation from IRCoT/ReAct:** We clarified that while IRCoT is recall-accumulative (often gathering noise), PRISM employs a novel "Prune-and-Recover" loop. The Selector (precision focused) aggressively filters distractors, while the Adder (recall focussed) specifically audits discarded evidence to recover missing links. This explicit, opposing-objective collaboration is structurally distinct from standard tool-use or accumulation frameworks.

- **Comparison to ARM:** We clarified that the suggested missing baseline (ARM) focuses on aligning structured data, whereas PRISM addresses multi-hop reasoning over unstructured text, making our comparisons to IRCoT and SetR the most methodologically relevant.

---

 3. **Response to Reviewer T7mf**

Reviewer T7mf acknowledged the **clear introduction of modules** but questioned the novelty and specifically requested an **"ablation of the Adder"**.

- **Clarification on Ablation (Table 6):** We pointed out that Table 6 already contains the requested ablation under the label "w/o Adder" (which represents a single-pass selection without the loop). The "Full Model" achieves 90.9 recall, whereas the "w/o Adder" baseline achieves only 79.7 recall on HotpotQA. This 11.2% gap in recall explicitly quantifies the **Adder's contribution**. Because the Selector is precision-focused (conservative), the Adder is the sole component responsible for recovering that missing evidence to boost recall. We have updated the manuscript to avoid confusion.

- **Efficiency:** We addressed the "costly design" concern with the new Latency Analysis showing PRISM is actually faster than the primary baseline (IRCoT) in **Table 14**.

- **Clarification on Novelty & Baselines:** We emphasized that PRISM's asymmetric loop - where the Selector prunes distractors to mitigate the "lost-in-the-middle" phenomenon and the Adder audits the discarded pile for completeness, is structurally distinct from prior accumulative systems. We highlighted that this unique dynamic drives the high efficiency reported in **Table 14**, achieving a 73.3% context reduction and significantly lower latency (8.6s vs 16s for IRCoT). Furthermore, we incorporated the baselines requested by the reviewer into **Table 5**, demonstrating that PRISM significantly outperforms context compression methods like RECOMP and general ranking methods such as QD-RAG and RankRAG. We have also updated our Related Work section to include this reference and clarify the distinction with our method.

---

We have revised the manuscript (changes highlighted in **blue text**) based on the reviewers comments and we have added all the new experimental evidences in the paper. We hope this summary assists in your final assessment. If there are any remaining concerns or further suggestions for improvement, please do not hesitate to let us know. We are willing to provide further details or conduct additional experiments to ensure the paper meets your expectations.

Thank you for your time and consideration.

Sincerely,

Authors

---

### Note · Authors · 2026-01-05

**Comment:**

After careful consideration of the reviews, we have decided to withdraw this submission.

Thank you.

Sincerely,

Authors

**Withdrawal Confirmation:**

I have read and agree with the venue's withdrawal policy on behalf of myself and my co-authors.